# Effects of Composting Yard Waste Temperature on Seed Germination of a Major Tropical Invasive Weed, *Leucaena leucocephala*

**Min Pan** [1,*], **Ling Chui Hui** [2], **Caroline Man Yee Law** [2] **and Sen Mei Auyeung** [2]

1 Department of Applied Science, School of Science and Technology, Hong Kong Metropolitan University, Ho Man Tin, Kowloon, Hong Kong SAR, China
2 Faculty of Design and Environment, Technological and Higher Education Institute of Hong Kong, Hong Kong SAR, China
* Correspondence: mpan@hkmu.edu.hk; Tel.: +86-852-312-026-34

**Abstract:** Composting is an environmental-friendly option for yard waste management, and produces products for improving soil quality. However, there is a weed dispersal risk if the compost contains many active weed seeds. This study assessed the potential of composting in minimizing the seed germination of a major tropical invasive weed, *Leucaena leucocephala*. The germination of the species was tested after two different sets of thermal treatments, i.e., (1) different constant temperatures (20 °C, 30 °C, 40 °C, 50 °C, 60 °C, and 70 °C) for 5 days, and (2) composting temperature (simulating the temperature profile of a typical composting process) for 60 days. A three-month growth test was further conducted for the seeds treated with the composting temperature. The seeds were present either alone (N-seeds) or mixed with wood chips (W-seeds) when thermally treated. A constant temperature treatment of 70 °C suppressed the seed germination to a low rate. For the composting temperature treatment, the germination percentage of the N-seeds and W-seeds were reduced from around 60% to 22.7% and 12.7%, respectively. This preliminary study suggested that the temperature should reach as high as 70 °C in the composting process to guarantee the suppression of the germination of the seeds of *L. leucocephala*, particularly when the seeds are contained within seed pods during composting.

**Keywords:** yard waste; compost; *Leucaena leucocephala*; invasive species; seed germination; temperature effect

## 1. Introduction

Yard waste (green or garden waste) refers to all different types of biodegradable vegetation waste matters. Some typical yard waste types include grass clippings, leaves, branches, and tree trunks. With the growing demand for urban greening, it is expected that the amount of yard waste will rise continuously with the increasing amount of planting [1]. To minimize the landfill disposal of yard waste and the associated environmental problems such as methane gas production [2], recycling or reusing yard waste are alternative sustainable options. For example, plant litter can be used as mulch onsite directly [3], or yard waste can be a source of biomass for energy production after processing [4]. Composting yard waste is another popular option for sustainable yard waste management. Composting is a process of controlled biodegradation of organic matter, and finally results in a stable product, which is the "compost".

Yard waste composts are valuable organic soil amendments with a high potential to improve various soil properties [5,6]. However, the weed seeds in the yard waste may create a risk of dispersal of local weeds and invasive species [7], particularly for the yard waste from weed species. During the composting process, organic materials are degraded by micro-organisms, and heat is released because of microbial activity. Though it

is suggested that almost all weed seeds are killed by the high temperature (55–65 °C) of a typical composting process, the effects are indeed species-dependent [8–10].

*Leucaena leucocephala* is one of the world's 100 worst invasive alien species [11] and is commonly found in tropical regions. It is a perennial broad-leaved plant species from the Fabaceae family and a multipurpose species. The different parts of the species are of valuable use and have a wide range of applications [12]. Nevertheless, its high invasiveness poses a great environmental risk outside its native range [13]. The high aggressiveness of the species is due to a number of attributes, including massive seed production, hard seed coat, high resprout ability, and its allelopathic property, which gives the species a high competing capability [13–16]. Particularly, the hard seed coat of *L. leucocephala* enables a persistent seedbank through offering strong protection from environmental stresses. The hardseededness is rather common among the members of Fabaceae [17,18].

It is reported that *L. leucocephala* has brought negative ecological consequences in different regions over the world, such as hindering reforestation, affecting vegetation production, and threatening native habitats [19]. Moreover, the species is regarded as invasive in over 100 countries according to the database of CABI [20]. To minimize the spread and negative impacts of *L. leucocephala*, mechanical control through cutting is sometimes carried out. Flowering and fruiting of *L. leucocephala* occur all around the year. Therefore, removal time plays a small role in minimizing the number of weed seeds contained in the yard waste. As the yard waste of *L. leucocephala* unavoidably comes with a large amount of the seeds, the yard waste must be carefully treated for further use. Though composting the yard waste may kill the weed seeds, currently, there are no studies that have assessed the effect of the high temperature of the composting process on this invasive species in tropical regions. Past studies were either focused on the effect of temperature on enhancing the germination [17,21–24], or on the inhibition of the germination by hot water in a short period [25]. In addition, many previous studies related to the inhibition of seed weed germination were in the context of composting manure (e.g., [26–29]). Research evaluating the topic against yard waste mass as a base material is still lacking.

Consequently, this research aims to act as a preliminary study to provide knowledge regarding the effects of the temperature of the composting process on inhibiting the seed germination of *L. leucocephala*. The specific objectives of the study were (1) to assess the effects of the temperature of the composting process on the seed germination and short-term growth performance of *L. leucocephala*, and (2) to discuss the effectiveness of composting yard waste containing weed seeds of *L. leucocephala* in minimizing weed dispersal risk.

## 2. Materials and Methods

### 2.1. Seeds of Leucaena leucocephala and Wood Chips

*L. leucocephala* is commonly found and regarded as an undesirable species in Hong Kong. The removal of this invasive species is frequently conducted to minimize the potential risks brought. The 100 seeds of *L. leucocephala* and 5 kg wood chips used in the experiment were provided by Eco Greentech Limited, Hong Kong. Matured seeds were collected during the tree removal works of *L. leucocephala* in 2021. The seeds and wood chips were kept in a 4 °C refrigerator before the experiment to preserve the materials, particularly the seeds, through slowing down the cellular chemistry. The seed viability was tested according to the International Seed Testing Association protocol [30].

### 2.2. Thermal Treatments on Seeds

The selected mature seeds of *L. leucocephala* were subjected to two different types of thermal treatments, namely constant temperature treatment and composting temperature treatment. In the constant temperature treatment, the seeds were exposed to a set constant temperature for 5 days. In the composting temperature treatment, the seeds were exposed to a temperature profile of a typical composting process for 60 days. Two sets of seeds were tested, including seeds that were present alone (N-seeds), and seeds that were mixed with wood chips (W-seeds). The N-seeds simulated the seeds that are within the pods

during composting and, therefore, somehow isolated from the moisture condition or other environmental factors of the composting process. Alternatively, the W-seeds imitated seeds that are released from pods and directly in contact with the larger yard waste mass. For each treatment group, 100 N-seeds and 100 W-seeds (100 seeds mixed with 200 g wood chips) were used.

### 2.2.1. Constant Temperature Treatment

Six different temperatures points from 20 °C to 70 °C, at an interval of ten degrees Celsius, i.e., 20 °C, 30 °C, 40 °C, 50 °C, 60 °C, and 70 °C, were set. The temperature range was referred to the range in a typical composting process. For each temperature point, 100 N-seeds and 100 W-seeds were heated in an oven (DSO-3000 DF, Taiwan), with the constant temperature maintained for 5 days. The seeds were then subjected to a germination test.

### 2.2.2. Composting Temperature Treatment

One hundred W-seeds and N-seeds were placed in an oven with set temperature changes simulating a typical composting process (Figure 1). The temperature profile was referred to the temperature profile of the composting treatment used by a local yard waste recycling company (Eco Greentech Limited, Hong Kong, China), which used a typical composting method. A typical composting process is characterized by four different phases based on the temperatures, namely the mesophilic phase; thermophilic phase; second mesophilic phase; and, lastly, maturation/curing phase. The temperature of the compost can reach >65 °C [31] and maintain at >55 °C for a certain period [7] to destroy pathogens and weeds. The whole treatment lasted for 60 days, with the first 30 days as the active phase and the remaining 30 days as the curing phase. The settings of the temperature profile were as follows (Figure 1): the first 8 days (day 0 to day 8) were the mesophilic phase, and the temperatures increased from 24 °C to 40 °C. The next phase was the thermophilic phase that lasted for 22 days (day 8 to day 30). The temperature in this phase started from 40 °C and increased rapidly to reach 65 °C on day 21, and then decreased gradually to 54 °C in the remaining 9 days. The next 13 days (day 30 to day 43) were the mesophilic phase. At this phase, the temperatures decreased quickly from 54 °C to 21 °C. In the last 17 days (day 43 to day 60), which was the maturation phase, the temperatures were maintained at 21 °C. A germination test was performed for the treated seeds later. One hundred seeds that did not receive the thermal treatments were included as a control.

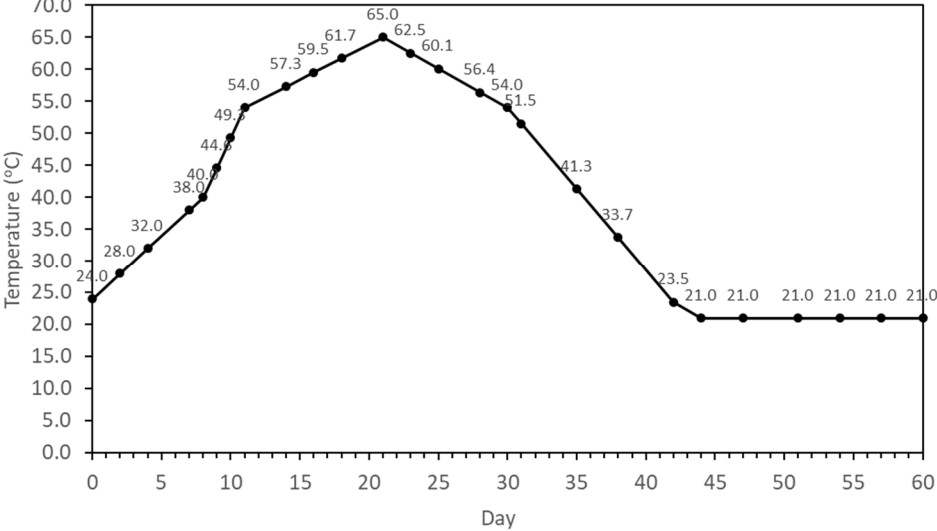

**Figure 1.** The temperature profile of the composting temperature treatment in the study.

### 2.3. Germination Test and Seedling Growth Test

2.3.1. Germination Test

The seed germination test followed the OECD procedure [32]. All seeds were first surface-sterilized in sodium hypochlorite solution (14%) for 10 min, and rinsed with deionized water afterwards. Five milliliters of deionized (DI) water was added to a 10 cm Petri dish with filter paper. Twenty seeds of *L. leucocephala* were placed on a filter paper in accordance with the ASTM standard germination protocol [33]. The Petri dishes were covered with lids before being kept in an incubator (JSPC-300C, growth chamber). The environment of the incubator was maintained at 25 ± 0.5 °C, 80% humidity, and in total darkness. Water loss in the Petri dishes was monitored every day by weighing, and distilled water was added if necessary. A seed was counted as germinated if the radicle was over 2.0 mm in length. The seed germination percentage and the radicle and plumule lengths were measured after 7 days of incubation to assess the inhibition effects. There were three replicates for each treatment group.

2.3.2. Seedling Growth Test

The seedling growth test was only conducted for the germinated seeds treated with the composting temperature treatment. This was because this treatment reflected the actual conditions of temperature variations during composting. The seedling growth test was conducted in a greenhouse with controlled temperature (25 ± 2 °C) and humidity (70%) for three months. There were three replicates for each treatment group. The planting procedure followed Chow and Pan [34]. Briefly, sand, peat moss, and pond clay were mixed to prepare the sandy loam soil, with 78.4% sand, 15.4% silt, and 6.2% clay. The pH of the original soil was 6.8 ± 0.3 and the organic carbon was at 0.62%. Each pot contained 3 kg potting mixes and was irrigated with 200 mL water every two days. No additional fertilizers were added during the growing period to avoid any potential influences on the results. The seedling height was recorded every week during the three months to measure the growth performance. The soil moisture content and soil temperature were monitored at a weekly interval. They were found to have remained steady throughout the experimental period.

Soil samples were extracted from the pots at the end of the seedling growth test. The samples were ground into powder and sieved through a 2-mm sieve. The procedures of the soil analysis followed that outlined in a previous study [34]. The measured soil physicochemical properties included pH value, conductivity, total organic carbon (TOC), and water-holding capacity. A pH meter with deionized water (1:1) was used to determine the soil pH value. TOC was tested through a TOC analyzer (Shimadzu, Kyoto, Japan). The maximum water-holding capacity was measured following the protocol of the International Organization for Standardization (ISO, 2012). The extraction method by the micro-Kjeldahl method was applied for the measurement of nutrients, including the total N, P, and K contents. The total N content extracted was examined by continuous flow analysis (CFA), whereas the total P and K contents were measured by atomic absorption spectrometry (AAS, Perkin Elmer 4100 ZL, Waltham, MA, USA).

### 2.4. Statistical Analysis

Data analysis was performed using the functions available in the Base R. Data were subjected to the analysis of variance (one-way ANOVA) and post-hoc Tukey HSD test to compare the effects of different treatments. The Shapiro–Wilk test and Levene's test were used to check the normality and the equal variances of the data. The Kruskal–Wallis H Test with Bonferroni correction would be applied if the data were not normal. A three-parameter Gaussian model was used to analyze the relationship of temperatures under constant temperature treatments with the germination percentage and the growth of the radicle and plumule at the germination stage [35]. The model was fit by using non-linear least squares (nls function in R) with the Gaussian function: $G = a \times e^{\left[-0.5 \times \{(x - b)/c\}^2\right]}$.

The "modelr" package in R was applied to calculate the pseudo-$R^2$ of the models. A *p*-value < 0.05 indicated a significant difference.

## 3. Results and Discussion

### 3.1. Effect of Constant Temperature Treatment on Seed Germination of L. leucocephala

Overall, the variations of germination percentage, as well as radicle and plumule growth, among the seeds receiving constant temperature treatments followed a very similar pattern along with the different temperature levels (Figure 2). The relationships of these three dependent variables and the treatment temperature were well fitted to the Gaussian models, forming the bell curves. The germination percentage, radicle length, and plumule length all increased as the temperature rose initially and began to decrease as the treatment temperature increased. For the W-seeds, it was predicted that the germination percentage, radicle length, and plumule length would start to decline when the temperature treatments were over 37.9 °C, 40.0 °C, and 38.9 °C, respectively. The corresponding temperature thresholds for N-seeds were 40.6 °C, 42.3 °C, and 40.7 °C, which were all slightly higher than that of the W-seeds. Though the W-seeds and N-seeds showed a similar trend of variations for all the three germination parameters, the W-seeds generally had a better germination performance at a lower temperature. This trend was absent or even reversed at a higher temperature. The N-seeds tended to be similarly or even less suppressed when the treatment temperature was higher.

A constant temperature treatment at 70 °C for five days was observed to suppress the seed germination to a great extent, but not a complete suppression. At 70 °C treatment, the observed germination percentages were 1.0% for W-seeds and 3.0% for N-seeds. In the Gaussian models, the predicted germination percentage was higher than the observed one at 70 °C, which was 5.1% for W-seeds and 7.5% for N-seeds. The predicted and observed radicle lengths of the W-seeds were 18.5 mm and 16.3 mm, and that of the N-seeds were 24.5 mm and 22.3 mm. The predicted and observed plumule length of the W-seeds were 8.2 mm and 12.6 mm, and that of the N-seeds were 9.6 mm and 15.5 mm.

As with the other plant species, temperature is one crucial factor that influences the seed germination of *L. leucocephala*. There are minimum, optimum, and ceiling temperatures for the germination of seeds [36]. A proper temperature range promotes the overcoming of seed dormancy [37]. The temperature affects the germination both chemically and physically. Chemically, it regulates the production and sensitivity of hormones such as abscisic acid (ABA) and gibberellins (GA) to alter the germination percentage [38,39]. Heat induces the physical crack of the hard seed coat and, hence, facilitates the imbibition process of seeds by enabling water and oxygen to enter the embryo to overcome dormancy and trigger germination [24,40,41]. The inhibition on the seed germination and seeding vigor did not appear until the thermal treatment was at around 40 °C in our study. At the lower temperature range, the increasing temperatures favored the seed germination before reaching a sublethal temperature. When the temperature reached a threshold, which was around 40 °C in our experiment, the embryos started to be damaged, and the proteins were denatured. Therefore, the germination and plant growth were inhibited, and the degree of suppression was increased with increasing temperature. Hwang et al. [42] suggested that the optimum germination temperature for the species was treating the seeds at 35 °C, and the seed germination percentage started to decline at 40 °C, which was lower than our study. However, the experimental design of that study was different in that the temperature was the germination condition of the seeds. In our experiment, the temperature was the treatment condition of the seeds before germination. Therefore, the results are not comparable.

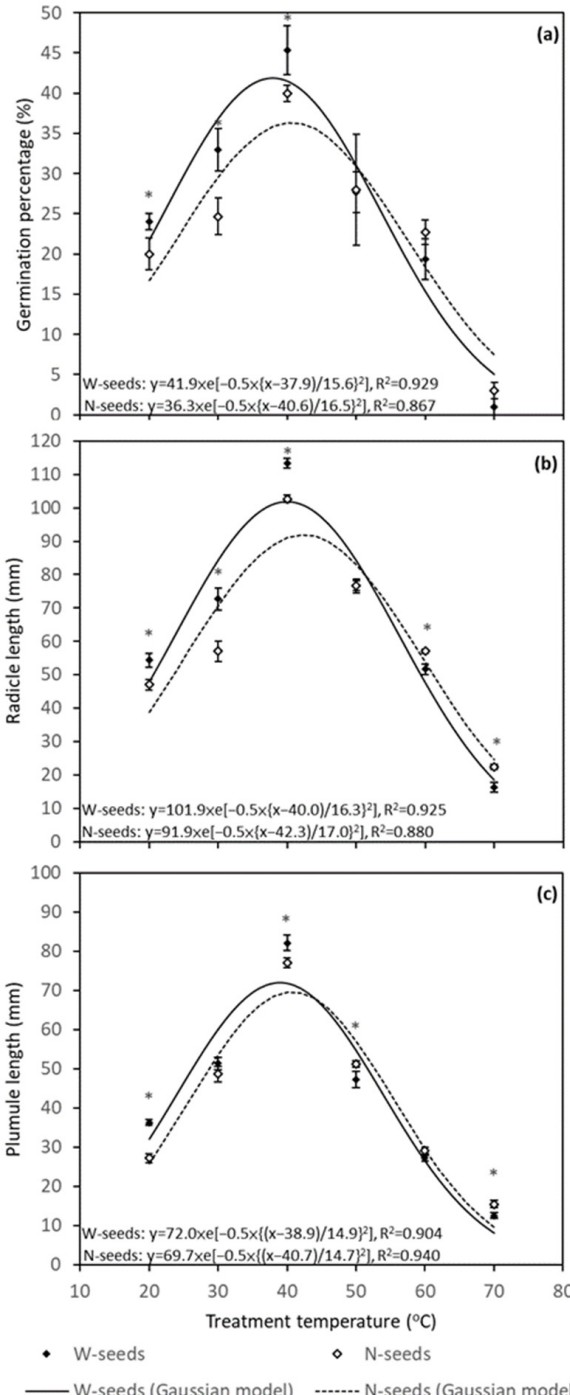

**Figure 2.** The (**a**) germination percentage, (**b**) radicle length, and (**c**) plumule length of the constant-temperature-treated seeds of *Leucaena leucocephala* mixed with wood chips (W-seeds) and present alone (N-seeds) after a 7-day germination test at six different treatment temperatures. * denotes a significant difference within a constant temperature.

It is suggested that the critical temperatures to suppress germination are in the range of 50–80 °C [8]. According to our findings, only a temperature as high as 70 °C was effective to inactivate or kill most, but not all, of the seeds, no matter if they were W-seeds or N-seeds. The seeds of *L. leucocephala* were rather tolerant to high temperature, probably due to the hard seed coat [43,44]. It was suggested that the high temperature tolerance of seeds is a critical factor for a species to be able to colonize in tropical regions where the grounds are

of very high temperature [45]. Therefore, the strong weedy characteristics of *L. leucocephala* may imply the high temperature tolerance of the seeds of this species.

The positive effects of the presence of wood chips on seed germination at the lower temperature range (below 40 °C) may be due to the trapping of heat and moisture by the wood chips. The slightly warmer temperature experienced by the seeds and the smaller loss of water content favored the seed germination. However, the trapping of heat became a negative factor at a higher temperature range (60 °C or above). This was because the seeds experienced an even higher temperature at a range that was detrimental to them. In addition, the moist condition lowered the resistivity of the seeds to high temperature [26].

### 3.2. Effect of Composting Temperature Treatment on Seed Germination of L. leucocephala

The germination percentage of *L. Leucocephala* was highly suppressed by the composting temperature treatment (Figure 3). The inhibition effect on the W-seeds was more prominent compared with the N-seeds. Specifically, the treatment reduced the germination percentage of W-seeds from 59.2% to 12.7% (close to five-fold), compared with that of N-seeds to 21.7% (around three-fold). A similar suppression pattern was observed for radicle and plumule elongation. The W-seeds had a more significant reduction in radicle and plumule length than the N-seeds compared with the control seeds. A respective 44.8% and 25.2% reduction in radicle length, and 51.1% and 22.7% reduction in plumule length were observed for the W-seeds (radicle length: 66.0 mm; plumule length: 36.7 mm) and the N-seeds (radicle length: 89.3 mm; plumule length: 58.0 mm).

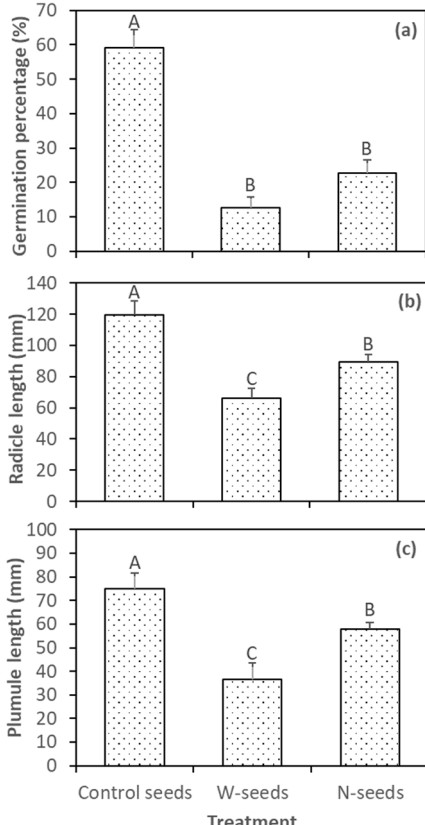

**Figure 3.** The (**a**) germination percentage, (**b**) radicle length, and (**c**) plumule length of the control seeds, composting-temperature-treated seeds mixed with wood chips (W-seeds), and composting-temperature-treated seeds present alone (N-seeds) of *Leucaena leucocephala* after a 7-day germination test. Different letters denote significant differences across different treatments.

Liu et al. [29] studied the effects of composting on the weed seeds of eight species. They identified that the seeds of seven of them were inactivated after being in the mesophilic

phase for two days. However, the seeds of field bindweed remained active throughout the phase. Similar to *L. leucocephala*, the seed of field bindweed has a thick and hard seed coat, and protects the embryo of the seed from extreme environments. Another example is from the study of Eghball and Lesoing [26]. They found that some velvetleaf (*Abutilon theophrasti*) seeds still survived after a composting process that reached 60 °C, whereas the seeds of all other studied species completely lost viability. Again, the seeds of velvetleaf have hard seed coats [46]. These suggested that the temperature of the composting process must reach a high point to effectively suppress seeds with high temperature tolerance, such as the ones with hard seed coats.

The highest temperature point was 65 °C in the composting temperature treatment in our study. The lower germination percentage in the constant temperature treatment at 70 °C, compared with the composting temperature treatment, revealed that the temperature level may be more important than the temperature duration in inhibiting germination. This is in accordance with the suggestion of Thompson et al. [8]. A prolonged composting temperature treatment with a lower maximum temperature produced a poorer inhibition effect when compared with exposing the seeds to a more extreme temperature for a shorter period. Indeed, the seed germination percentage in the composting temperature treatment (12.7% and 22.7%) was lower for W-seeds or comparable for N-seeds with that in the constant temperature treatment at 60 °C (19.3% and 22.7%). It was proposed in different studies that the thermal treatment of seeds at around or lower than 60 °C for a few days is sufficient to suppress seed germination completely. For example, Nishida et al. [47] recommended that a thermal treatment of over 60 °C for more than around 6 days can kill the seeds in compost. Grundy et al. [7] suggested that exposing seeds to 55 °C for around three days is adequate to eliminate seed germination. However, the complete suppression of seed germination was not achieved in our experimental conditions, indicating that temperature alone is not the only contributing factor to produce the suppression effect at this temperature range. This is in accordance with the view of Larney and Blackshaw [9]. They argued that the relationship between temperature and seed viability was not definitive. Their regression models demonstrated that the temperature only accounted for <30% of the variation of seed viability. In our study, it was chiefly the temperature condition of the composting process being simulated. However, there are many other factors that also contribute to the killing of weed seeds during composting, such as the release of phytotoxins in leachates from compost mixtures [28,29,48]. Moreover, different previous studies have showed that the moisture content largely affects the effect of temperature [26,37,49]. Under moist conditions, the temperature required to kill weed seeds is comparatively lower. Therefore, it is likely that under a typical compost condition, the seed germination percentage will be lower than the one observed in our study. The moisture condition and release of substances during a real composting process are probably more favorable for inhibiting seed germination.

### 3.3. Effect of Composting Temperature Treatment on Seedling Growth of L. leucocephala

Figure 4 displays the trend of the seedling growth from the seeds with and without the composting temperature treatment. In the first week, the height of the seedlings from the control seeds (53.5 ± 4.6 mm) was significantly greater than that of the W-seeds (43.0 ± 3.5 mm), whereas N-seeds had an even smaller height (23.9 ± 3.4 mm). The height of the seedlings from the W-seeds and N-seeds was 19.6% and 55.5% lower than that of the control seeds, respectively. The differences were narrowed as the planting days increased. The final height of the seedlings from the N-seeds (102.6 ± 3.9 mm) was similar to that from the control seeds (104.6 ± 13.5 mm). For the seedlings from the W-seeds, a lower final height (90.0 ± 2.8 mm) was resulted compared with the seedlings from the control seeds. Yet, the difference was not large (−14.0%) and not significant. Overall, the seedlings from the W-seeds exhibited a 280.6% increase in height from week 1 to week 12, followed by the N-seeds (139.9%) and control seeds (95.1%). The negative effect of the composting temperature treatment on the seedling growth of the germinated seeds, in terms of height, was lessened or even disappeared after a certain growing period.

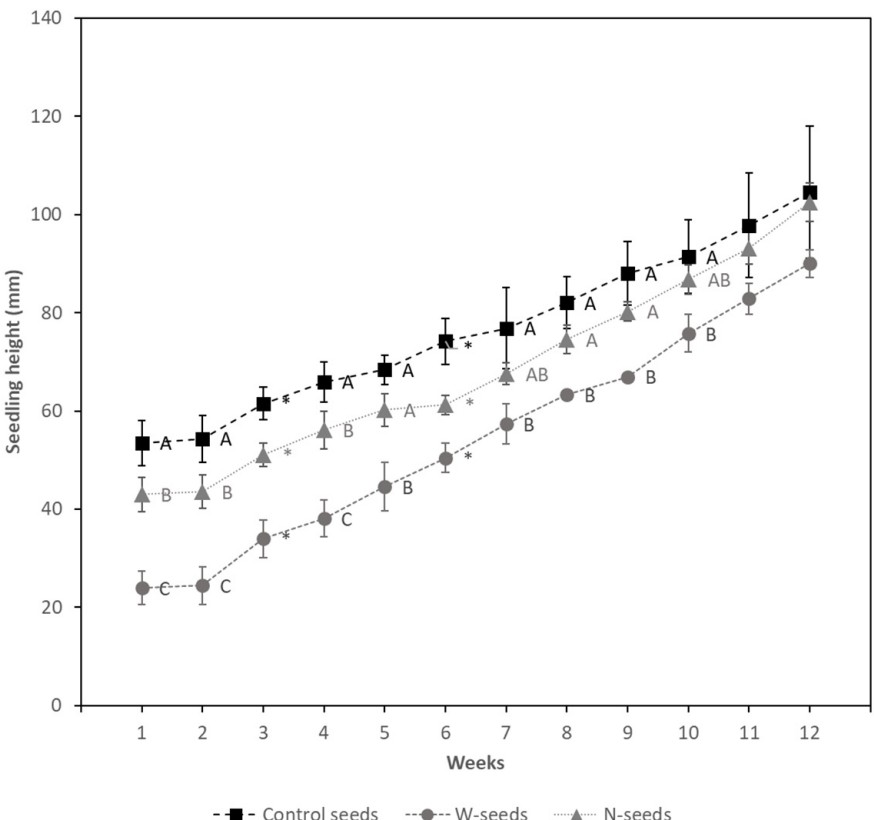

**Figure 4.** The height of the *Leucaena leucocephala* seedlings from the control seeds, composting-temperature-treated seeds mixed with wood chips (W-seeds), and composting-temperature-treated seeds present alone (N-seeds) throughout a three-month (12 weeks) growing period. Different letters denote significant differences in height across different treatments in the same week. * denotes a significant difference of height detected by the Kruskal–Wallis H test, but not by post-hoc pairwise comparison, at weeks 3 and 6, in which the data were not normal.

The soil properties did not show any significant differences at the end of the seedling growth test (Table 1). However, there was a rather high total N content and P content in the soils of the W-seeds, which were almost 40% and 50% higher than that of the control seeds, respectively. As the initial nutrient contents were supposed to be similar, the higher final nutrient contents likely indicated that the nutrient absorption was lower, and, hence, the seedling growth. This hints at the weaker growth of seedlings from W-seeds at the early stage of growth. Nevertheless, the lack of significant differences between the nutrient values echoes the findings that the growth of seedlings from W-seeds, N-seeds, and control seeds did not differ markedly after growing for some time.

**Table 1.** Mean (M) and standard deviation (SD) values of the soil physical and chemical properties at the end of the seedling growth test of the control seeds, composting-temperature-treated seeds mixed with wood chips (W-seeds), and composting-temperature-treated seeds present alone (N-seeds) of *Leucaena leucocephala*.

| Treatment | Available N (mg/kg) | | Available P (mg/kg) | | Available K (mg/kg) | | Total N (g/kg) | | Total P (g/kg) | | Total K (g/kg) | | pH | | Conductivity (mS/cm) | | Water Holding Capacity (%) | | TOC (%) | |
|---|---|---|---|---|---|---|---|---|---|---|---|---|---|---|---|---|---|---|---|---|
| | M | SD | M | SD | M | SD | M | SD | M | SD | M | SD | M | SD | M | SD | M | SD | M | SD |
| Control seeds | 76.1 | 8.0 | 92.9 | 5.0 | 73.7 | 14.7 | 5.5 | 0.2 | 5.1 | 0.3 | 66.0 | 2.6 | 37.0 | 3.4 | 98.6 | 8.5 | 172.5 | 33.2 | 283.4 | 61.4 |
| W-seeds | 77.5 | 11.6 | 89.4 | 10.6 | 77.8 | 15.9 | 5.4 | 0.3 | 4.9 | 0.6 | 63.0 | 4.0 | 36.3 | 2.9 | 137.1 | 25.9 | 255.4 | 51.2 | 280.6 | 21.7 |
| N-seeds | 82.3 | 6.6 | 83.3 | 3.7 | 81.8 | 10.0 | 5.5 | 0.1 | 5.6 | 0.4 | 65.3 | 5.5 | 34.1 | 4.0 | 116.5 | 13.3 | 172.7 | 41.1 | 258.5 | 50.2 |

Almost all studies that evaluated the effect of the temperature of composting on seed germination did not measure the growth of seeds in the germination stage or over a longer term of growth. One possible reason is that the ultimate goal should be the complete elimination of seed germination, rather than the suppression of seedling growth. Yet, evaluating the growth performance of seeds after germination can provide additional information on the competitiveness and long-term survival of the plants affected by the composting process. Our findings demonstrated that the seeds that survive the harsh condition of the high temperature of composting probably show comparative growth to the untreated seeds in the long term. The heat might only provide a temporary suppression effect, but does not bring irreversible damage to the embryo of the seeds. The seeds may be in a stage called thermoinhibition at first, and, therefore, take a longer time to germinate [50]. It was suggested that the temporary inhibition was possibly caused by different factors, such as changes in embryo coverings, hormone levels, protein products, etc. However, the influences are mainly at the early stage of growth, and likely to become negligible in the long term, as suggested by our findings. Accordingly, a failure to eliminate the germination ability of the seed weeds in compost is risky.

The lower germination percentage and poorer early growth performance of the W-seeds in the composting temperature treatment can again be attributed to the trapping of heat and moisture by the wood chips, which established a more stressful environment for seeds, as discussed in the previous section. Lastly, it was found in our study that the germination percentage ($59.2 \pm 5.2\%$) of the control seeds was higher than that of those that had received the constant temperature treatment at the optimal temperature of 40 °C ($45.3 \pm 3.1\%$). This was also observed in the study of Thompson et al. [8], but no explanation was given for the phenomenon. The lower ambient relative humidity in the oven environment probably led to a lower germination percentage of the thermal-treated seeds. Further studies can be conducted to assess the seed germination percentage of the species with the moisture effects or other factors, or in an authentic composting process.

### 3.4. Practical Implications

An effective composting process that can produce weed-seed-free compost provides an alternative option for yard waste management, especially for yard waste containing a high quantity of weed seeds. It offers useful products for agricultural or horticultural purposes. However, a failure to produce weed-seed-free compost can lead to devastating consequences for ecosystems due to the high invasiveness of some weed species such as *L. leucocephala*. As seen from our results, the seeds that survived the high temperature of composting (as high as 65 °C) still showed considerable growth after germination. The suppression of growth was only evidenced at the early stage of seedling growth. Therefore, the risk is still present if the seed germination cannot be completely hindered. The application of compost with even a very small amount of germinable seeds will lead to weed dispersal, and, therefore, is still undesirable.

The high temperature tolerance of the seeds of *L. leucocephala*, as revealed in our study, suggested that a high temperature must be achieved during composting to eliminate the risk of seed dispersal. A peak temperature of 70 °C is desirable to balance the need for weed seed elimination and the quality of compost. A temperature higher than this level is often deemed undesirable for composting, as it will inhibit biodegradation by micro-organisms [51] and promote the release of ammonia [52].

The seeds mixed with wood chips (W-seeds) were close to the condition in which the seeds are released from pods and directly mixed with the other yard waste in the real composting process. The seeds that were present alone (N-seeds) were similar to those that are contained within seed pods. The effect of high temperature was found to be lessened for N-seeds. This implied that if seed pods, rather than individual seeds, are composted, the effect of composting on minimizing the seed dispersal risk will be impeded to a certain extent. Further, the seeds within the pods may be better protected from the other environmental factors that favor the inhibition of seed germination. However, to save

labor, it is a common practice that pods will be mixed with the other yard waste directly and composted together in the practical application. From this point of view, reaching a high temperature during composting is especially important.

Nevertheless, it is important to ensure the composting process can maintain a high temperature for a certain period, such as five days, to suppress the seed germination of *L. leucocephala*, a species with seeds with a high temperature tolerance. Reaching a temperature of 70 °C can be set as the target. Moreover, the avoidance of cool spots in compost by sufficient turning [7] is particularly important. Otherwise, some weed seeds may not be killed due to the insufficiently high temperature at certain locations of the compost.

## 4. Conclusions

This study was the first to investigate the effect of a high temperature during compost on inhibiting the seed germination of *L. leucocephala*, a highly invasive species in many tropical regions. The preliminary study revealed the high temperature tolerance of the seeds of the species. The inhibition effect did not appear until the temperature level of the constant thermal treatment reached around 40 °C. The thermal treatment of 70 °C for five days effectively suppressed the germination of the weed seeds to a very low level, but a low percentage of seeds still germinated. The 60-day composting temperature treatment that reached over 60 °C for several days led to substantial reduction, but not a complete elimination of seed germination. The temperature level was more critical than the duration of thermal treatment in suppressing seed germination. In addition, the N-seeds were less affected than the W-seeds by the thermal treatments at high temperature levels. For the seeds that survived the composting temperature treatment, the seedling growth was lower than that of the control seeds at the early growth stage. However, the differences tended to be lost after some time.

A temperature as high as 70 °C is required to effectively kill the weed seeds of the species. A high temperature of composting is particularly important when treating seeds contained within the seed pods, as the seeds are isolated from the other environmental factors of composting that contribute to the killing of weed seeds. The findings of our study can be used as a useful reference for designing a composting process in a large-scale composting plant setting to alleviate the risk of seed dispersal in this species.

**Author Contributions:** Conceptualization, M.P.; methodology, M.P. and S.M.A.; validation, M.P.; formal analysis, S.M.A. and L.C.H.; investigation, L.C.H.; resources, M.P.; data curation, M.P.; writing—original draft preparation, M.P., L.C.H. and S.M.A.; writing—review and editing, M.P. and C.M.Y.L.; visualization, L.C.H.; supervision, M.P. and C.M.Y.L.; project administration, M.P. and C.M.Y.L.; funding acquisition, M.P. and C.M.Y.L. All authors have read and agreed to the published version of the manuscript.

**Funding:** This research was funded by the research donation from Eco Greentech Limited and Research Grants Council of Hong Kong, grant numbers UGC/RMG/011, UGC/FDS25(16)/M02/19, and UGC/FDS25(16)/M01/20.

**Institutional Review Board Statement:** Not applicable.

**Informed Consent Statement:** Not applicable.

**Data Availability Statement:** The data sets used during the current study are available from the corresponding author on reasonable request.

**Acknowledgments:** We would like to convey sincere gratitude to Eco Greentech Limited for their donation support and providing seeds and wood chips for the experiment.

**Conflicts of Interest:** The authors declare no conflict of interest.

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
