# Peer review of "Effects of Composting Yard Waste Temperature on Seed Germination of a Major Tropical Invasive Weed, Leucaena leucocephala"

_sustainability, doi:10.3390/su142013638_

Round 1

Reviewer 1 Report

The manuscript describes the impact of different composting temperatures on seed germination of an invasive weed. The highest composting temperature significantly suppressed seed germination. The experiment is not designed by expert seed scientist; therefore, had several flaws which deserve serious attention.  Plenty of improvements are needed in the manuscript. I have uploaded the annotated PDF for detailed comments. In addition to the comments on file, I have following suggestions,

- Please improve terminologies for the mechanical management of target invasive species. Whether it is mowing? Cutting? Pruning or what? The argument that cutting/mowing residues leave significant amount of seeds raises several doubts on the opted management strategies. If the mowing time is after seed set then it could not be considered as a viable option. Please provide sound background on this to strengthen your argument.

- The introduction section needs a little bit extension. The extent of current invasion and mowing residues should be included

- The MM section has missing details for the important traits such as root and shoot growth rate.

-The data for root and shoot growth for two experiments are presented differently. For constant temperature it seems average, while for composting temperature weekly data is given. This is a major flaw that the experiments were not treated equally

- The data of constant temperature needs a different analysis as such data are not suitable for ANOVA

- Was seed dormancy tested before the initiation of seed germination experiment? This is really important to have exact suppression of seed germination. Similar is the case for seed viability. Was it tested before or after the experiment

- Data analysis section must contain more details on the normality and homogeneity of variance

- The petri must be Petri throughout the text as it is the name of scientist who invented this dish

-Results are too wordy and complex. Please reanalyze data and simplify the expressions

- The discussion should compare the similar studies focused on suppressing seed germination

- Please recommend some sound advices based on your obtained

Reviewer 2 Report

Title: Effects of composting yard waste temperature on seed germination of a
major tropical invasive weed, Leucaena leucocephala

The study was well designed, conducted and presented.  I consider that the manuscript may be submitted for publication in the journal after completing the minor corrections noted below:

 L.18: What is the composting temperature?

Lines 78 – 95: This corresponds to the Materials and methods section. Thermal treatments on seeds

L 104: Why maintained the seeds at 4ºC?

L122. Day = day

L140: Clarify What it does mean DI?

Round 2

Reviewer 1 Report

I have evaluated the revised manuscript where authors have mostly addressed my comments. However, I have some reseravtions on the language, particularly in the abstract section. I have suggested some changes ot improve it.

The abstract still lacks a concrete take-home message, although it is a preliminary study. Would author receommend composting of N or W seeds? The messages must not be hypothetical.

Germination rate is frequently used in the manuscript which is actually seed germination percentage.

Again conclusion must be precise and concrete.
